# Active Pattern Classification for Automatic Visual Exploration of Multi-Dimensional Data

**Jie Li** \*, **Huailian Tan and Wentao Huang**

College of Intelligence and Computing, Tianjin University, Tianjin 300350, China; tanhuailian@tju.edu.cn (H.T.); 2119216075@tju.edu.cn (W.H.)

\* Correspondence: jie.li@tju.edu.cn

**Abstract:** The practice of applying a classifier (called a pattern classifier and abbreviated as PC below) in a visual analysis system to identify patterns from interactively generated visualizations is gradually emerging. Demonstrated cases in existing works focus on ideal scenarios where the analyst can determine all the pattern types in advance without adjusting the classifier settings during the exploration process. However, in most real-world scenarios, analysts know nothing about data patterns before exploring the dataset and inevitably find novel patterns during the exploration. This difference makes the traditional classifier training and application mode less suitable. Analysts have to artificially determine whether each generated visualization contains new data patterns to adjust the classifier setting, thus affecting the automation of the data exploration. This paper proposes a novel PC-based data exploration approach. The core of the approach is an active-learning indicator for automatically identifying visualizations involving new pattern classes. Analysts thus can apply PCs to explore data while dynamically adjusting the PCs using these visualizations. We further propose a PC-based visualization framework that takes full advantage of the PC in terms of efficiency by allowing analysts to explore an exploring space, rather than a single visualization at a time. The results of the quantitative experiment and the performance of participants in the user study demonstrate the effectiveness and usability of the method.

**Keywords:** interactive data exploration; automatic visualization; neural network; machine learning; active learning; visual analytics





## 1. Introduction

Interactive data exploration (IDE) is designed to identify potential data patterns in a dataset [1,2]. Typically, analysts repeatedly conduct two steps, i.e., (1) querying the dataset to generate a visualization and (2) reading the visualization to determine whether patterns exist. Although there are many methods to automate (1) by recommending the optimal visualization technique to present target data [3,4], the subject of (2) is still human. Subjective pattern identification reduces the IDE efficiency, imposes high workloads on analysts, and may produce biased conclusions limited by analysts' cognitive capability and experiences.

A recently proposed idea, that each visualization is a snapshot taken by a virtual camera around the dataset [5–7], has inspired researchers to use image classifiers that are widely used in the classification of real-world objects to identify data patterns in visualizations [8,9]. A pattern classifier can classify an input visualization into a class representing a pattern type, as shown in Figure 1, thus transforming pattern identification into a classification task. We are confident in the performance of the PC, considering that deep learning-based classifiers outperform humans in distinguishing tiny visual differences among real-world objects that are more complex and irregular than information in visualizations. On the other hand, a PC can improve the efficiency and effectiveness of IDE. Pattern identification, which is frequently conducted in IDE and consumes many

analysts' energy, will be taken up by PCs. Thus, analysts can pay more attention to other high-level tasks, such as controlling the exploring direction and gaining insights into the identified patterns. PCs make AI models and humans do what they are good at in IDE, avoiding the subjective pattern identification problems mentioned above.

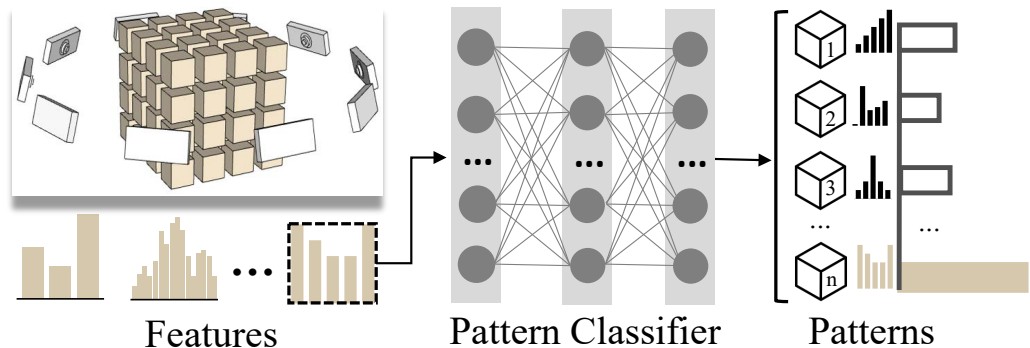

**Figure 1.** The basic idea of a pattern classifier, i.e., classifying an input visualization to a pattern class.

A pattern classifier differs from an image classifier significantly. Each image inherently belongs to a class. In contrast, pattern classes are artificial, and analysts know little about the patterns of a dataset before exploring it. As the analyst generates and reads visualizations, the number of recognized patterns gradually increases. Even though we can initialize a PC through manual sampling and labeling [10–13], the PC inevitably classifies a newly generated visualization that contains never-encountered patterns into an existing class during IDE.

This paper proposes a method for constructing and applying a PC in IDE. The method's core is an active-learning indicator, which can quantify how much a sample does not belong to any preset class (called an abnormal sample and a normal sample, vice versa). We propose an indicator based on an experiment phenomenon on classic classifiers; i.e., abnormal samples will produce more dramatic fluctuations than normal samples in classifiers' hidden layers. The indicator is computationally efficient and inherently normalized. Thus, analysts can (1) obtain a sample's indicator score before the PC assigns it to a class and (2) set a threshold to filter abnormal samples for updating the PC to cover all recognized pattern classes during IDE.

To construct, use and update pattern classifiers easily and efficiently in the face of unknown data, we further propose a PC-based IDE framework called random batch visualization exploration (RBVE). It enables interactively initializing and applying a PC in IDE and makes full use of the PC's advantages in exploration efficiency. Specifically, RBVE can quickly find representative patterns within analyst-specified exploring space by randomly synthesizing queries to generate a batch of visualizations and quickly identify their patterns through PCs. Such ability to explore the whole space at once is not available with traditional human-dominated IDE.

We conducted both objective and subjective experiments to evaluate the method's usability. Quantitative benchmarking against common active-learning indicators demonstrates that our indicator is more applicable in identifying abnormal samples. Furthermore, participants' better performance in completing pre-defined tasks and relatively positive comments further validate the applicability of the PC-based RBVE framework for optimizing the IDE process.

In short, we generalize the concept of a pattern classifier and give a method for applying pattern classifiers in practical IDE scenarios in which pattern classes change all the time. Our method includes the following two concrete contributions:

- We propose an indicator for automatically identifying abnormal samples, enabling dynamic updating of the pattern classifier to cover all identified patterns during IDE;

- We design an IDE framework called random batch visualization exploration, enabling analysts to find representative patterns of the whole exploring space with few interactions.

The remaining part of the paper is organized as follows. Section 2 reviews related works. After a brief description of the research problems in Section 3, we introduce the indicator and the visualization framework in Sections 4 and 5, respectively. Sections 6 and 7 describe the experiments. We conclude the paper in Section 8.

## 2. Related Work

We review two types of techniques relevant to our research, i.e., active sample recommendation and automatic pattern identification.

### 2.1. Active Sample Recommendation

Deep learning has developed rapidly and received widespread attention in the past decade. Obtaining sufficient training samples is a prerequisite for building any deep-learning model. Model developers, therefore, have to examine a large number of instances and assign each instance a label, which is time-consuming and costly.

Active learning [14] is an approach to recommending samples that are most likely to improve the model's prediction accuracy. This type of learning provides indicators to quantify samples' classification uncertainty. Model developers thus should select samples with higher uncertainty to label, which are more likely to be misclassified [15]. Many active-learning indicators are calculated based on the distribution of the classifier's penultimate layer (each value indicates the probability of a sample belonging to a class), such as least significant confidence [16], smallest margin [17], entropy-based sampling [18], Simpson diversity [19], etc. Another type of indicator follows the ensemble strategy, constructing multiple weak classifiers and selecting samples with the lowest consistency of their classification results in these weak classifiers. Representative multi-model based indicators include vote entropy [20,21] and Kullback–Leibler average [22]. Most active-learning indicators select samples that can significantly improve models' classification accuracy. In contrast, we need an indicator to reveal how likely a sample is to not belong to any existing class. None of the existing active-learning indicators are for this purpose.

Interactive labeling provides model developers with visual interfaces to identify, select, and label interesting instances to improve the labeling flexibility [10]. Many visualization tools integrate new workflows to reduce the samples to be labeled [23–26]. Arendt et al. [11] utilized the 1-nearest neighbor algorithm, which can automatically propagate the labels to unlabeled samples. Model developers thus only need to label a small number of samples. Beil et al. [12] proposed a cluster-clean-label workflow to iteratively optimize labeled clusters' purity. Bernard et al. [27] summarized a series of strategies for interactive labeling. However, relying solely on visual and interactive interfaces without any quantitative reference may produce sub-optimal sample selection [28].

Many methods combine the strengths of active learning and visualization to balance the efficiency and flexibility of labeling [29,30]. A common strategy is using active-learning indicators to select informative samples and visual interfaces to control the labeling process [28,31,32]. However, these methods target objects with fixed classes, and are inapplicable for the scenarios of pattern classification. Felix, Dasgupta, and Bertini [33] proposed exploratory labeling, which allows model developers to customize and refine labels over time. However, the labels depend on the developer-specified "seed terms", reflecting their subjective intent. Instead, our method can automatically identify samples with new pattern classes to update the classifier, avoiding unnecessary human intervention.

### 2.2. Automatic Pattern Identification

There are many indicators for detecting whether a visualization contains specific data patterns. The Scagnostics approach for detecting anomalies in SPLOM (scatterplot ma-

trix) [34,35] and subsequent improvements [36–41] are representative. Following the idea of Scagnostics, researchers have proposed many other indicators for a variety of visualization techniques, such as time series [42,43], treemap [44,45], parallel coordinates [46–50], parallel sets [51], star glyphs [52], and pixel-oriented displays [53]. Seo and Shneiderman [54] used ordinary statistics to select the most suitable views for showing filtered data. Berger and Hauser [55] extended Seo and Shneiderman's framework to rank dimensions according to indicators of subsets formed by brushing in coordinated views.

Visualization authoring can automatically generate visualizations that are likely to contain patterns according to the data distribution [56,57]. Many tools, such as SEEDB [58], Autovis [59], VisPilot [60], Foresight [61], DataShot [62], zenvisage [63], etc., also depend on indicators to generate visualizations that are likely to contain patterns. Important surveys of this type of research include [64–67]. Indicators, however, can only find specific pattern types [66]. In contrast, a PC can create a pattern class based on visualization examples generated on the fly and quickly discover the same class of visualizations from a large number of candidates.

Neural networks have also been used in authoring visualizations. Data2Vis [4] formulates visualization generation as a language translation problem, training an attention-based encoder–decoder network to generate Vega-lite specifications [68]. VizML [3] applies deep neural networks to predict design choices based on a corpus of existing visualizations (label) and the associated data sets (feature). DeepEye [69] uses a decision tree to evaluate the quality of visualizations. Draco [70] can find appropriate visual encodings using a learning-to-rank mode. A PC and visualization authoring focus on different stages of IDE and can be used together.

More and more researchers are using machine/deep learning-based techniques to improve the efficiency and effectiveness of data exploration [71–73]. Bosch et al. [74] and Snyder et al. [75] utilized classifiers to identify tweets relevant to analysis goals from the data stream. Heimerl et al. [76] designed a classifier-based text retrieval system. Gramazio, Huang, and Laidlaw [77] demonstrated the effectiveness of classic classifiers, such as KNN and SVM, in identifying meaningful data for common visual analysis tasks. However, these works treat classifiers as a preliminary data-filtering component, unlike our approach, which uses a classifier to mine data patterns directly.

Many works train machine-learning models to retrieve visualizations relevant to analyst-specified patterns. Law, Basole, and Wu [78] utilized a multinomial logistic regression model to determine whether two visualizations contain similar patterns. Dennig et al. [79] trained a model to rank similarity indicators reflecting analysts' preferences and used the best-ranked indicator to retrieve visualizations of interest. A PC does not require analysts to specify target patterns and therefore has better applicability.

Applying supervised classifiers to improve IDE automation is a newly emerging trend in visualization. Piet al. [9] built a classifier to identify four traffic congestion patterns from their proposed cumulative vehicle curves. However, since they determine pattern classes before training the classifier, they do not need to consider the challenge of identifying abnormal samples to update the classifier during IDE. Krueger et al. [8] proposed Facetto, a visualization tool that integrates a classifier to identify cancer and immune-cell types from microscopy images of human tumors and tissues. However, Facetto relies on unsupervised clustering and human interactions to identify new cell classes during IDE. In other words, although pattern identification is automatic by applying the classifier, the determination of pattern classes remains artificial. Our method automates the identification of new patterns from visualizations, enabling better use of the advantages of a PC in terms of exploration efficiency.

## 3. Problem Statement

We first introduce the application scenarios in which users use pattern classifiers for data exploration, based on which, the challenges of applying a PC in IDE are identified.

*3.1. Application Scenarios*

In a traditional data-analysis pipeline, analysts operate the system to generate the visualizations. Moreover, they observe the visualizations to determine the existence of any patterns. Subjective pattern exploration can lead to analysis bias and pose a significant workload for analysts.

Figure 2 illustrates the scenario of applying a pattern classifier for data analysis. The analyst interactively operates the visualization system to generate a visualization. The pattern classifier takes the generated visualization as input and automatically identifies patterns involved in the visualization by putting it into a specific pattern class. In this process, analysts do not need to perform tedious analytic tasks. Instead, they obtain patterns from the pattern classifier and only need to maintain the pattern classifier.

This approach reduces the burden of analysts because it frees users from the tedious task of reading visualizations and identifying patterns, allowing them to focus on higher-level tasks, such as controlling the exploration direction and adjusting the exploration process to understand the analysis results. The approach allows AI and humans to do what each is good at during data exploration, improving exploration efficiency and avoiding subjective bias.

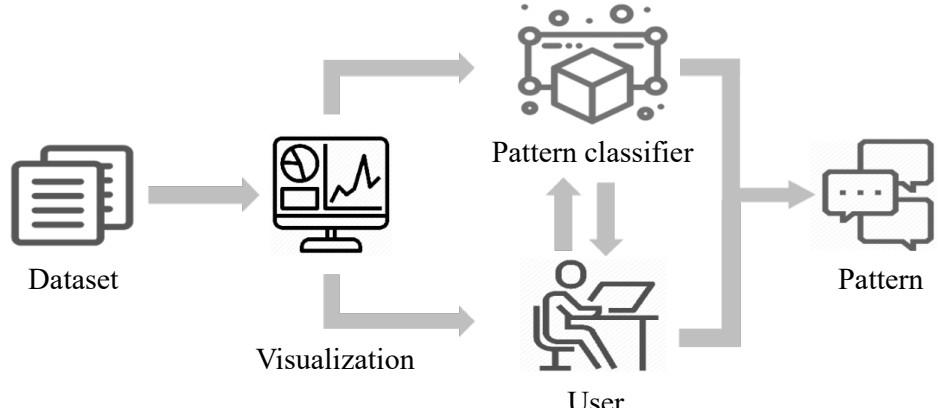

**Figure 2.** Scenarios for applying pattern classifiers for automated data analysis.

*3.2. Challenges*

The application of pattern classifiers for automated data analysis has the advantages of high efficiency and avoidance of subjective bias. However, there are two challenges to be solved in the application process, as follows:

- **C1**: Users may arbitrarily conduct data queries during IDE, inevitably generating visualizations with data patterns that have not been included in the PC. Thus, the first challenge to be resolved is how to automatically find these visualizations for updating PCs;
- **C2**: A PC changes the traditional IDE process taking the place of the human as the subject of pattern identification. How to construct a new visualization framework that adapts to this change and makes full use of a PC's advantage in exploration efficiency is the second challenge.

We will introduce an active-learning indicator for identifying abnormal samples (C1, Section 4) and a novel PC-based IDE framework that can efficiently discover representative patterns within a user-specified space (C2, Section 5).

## 4. Input Normality Measurement

We first introduce the concept of sub-model and then give the indicator according to the results of a preliminary study on classic image-classifier sub-models to distinguish between normal and abnormal samples.

### 4.1. Sub-Model

We define the sub-model according to the general classifier structure, which consists of an input layer, several hidden layers, and an output layer. Given a classifier with $N$ hidden layers, we can construct up to $N$ sub-models for it. Specifically, the $i$th sub-model should include the input layer, the first $i$ hidden layers of the original classifier, and a newly added output layer, as in Figure 3. The output layer of a sub-model has the same structure as the original classifier, but we need to train the output layer separately for each sub-model. Thus, its weights are different from those of the original classifier.

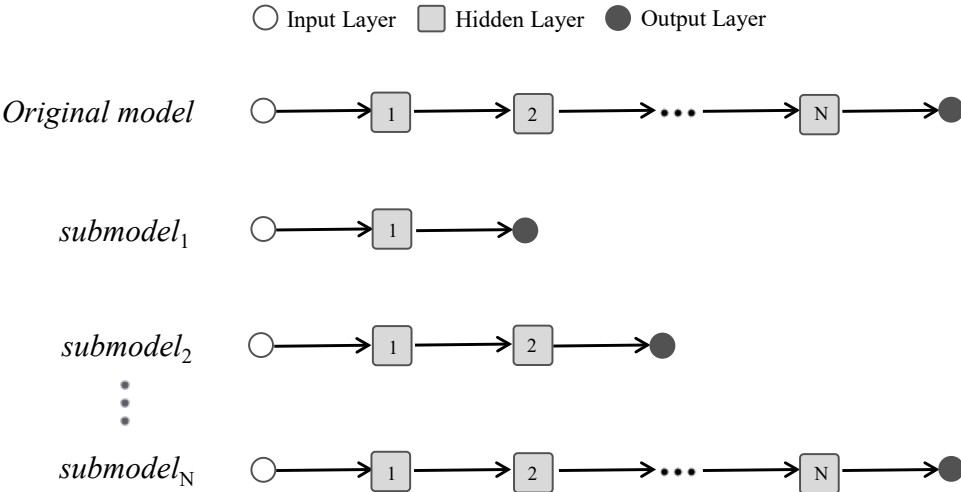

**Figure 3.** $N$ sub-models of a classifier with $N$ hidden layers

Each sub-model is an independent classifier with fewer hidden layers. Its outputs thus reflect the intermediate classification results of the original classifier. A sub-model takes the $i$th hidden layer of the original classifier as input and outputs classification results in the same form as the original classifier.

When training a sub-model, we only need to update its output layer, as it has the same input and hidden layers as the original classifier. We use the training set of the original classifier to train a sub-model. Let $(x, y)$ be a sample and $f_i(x)$ be the value of the $i$th hidden layer. We can take $(f_i(x), y)$ as a training sample of the $i$th sub-model. Therefore, each sub-model has the same number of training samples as the original classifier. Training sub-models can be efficient, as we only need to update a single output layer for each sub-model.

We propose two hypotheses about sub-models based on our understanding of neural networks. First, a classifier always has many hidden layers to improve its accuracy. Therefore, we think the outputs of sub-models will fluctuate randomly over the first few hidden layers and gradually converge to the same result as the original classifier. Second, classifying abnormal samples is more difficult. Their classification results fluctuate more significantly over multiple classes in the sub-model and they need to go through more hidden layers to reach a stable classification result.

According to the considerations, we give two hypotheses:

**Hypothesis 1 (H1).** *For both normal and abnormal samples, the closer the sub-models are to the input layer, the more significant the fluctuations in their classification results.*

**Hypothesis 2 (H2).** *Abnormal samples need to go through more hidden layers before their sub-models converge to the same classification results as the original classifier.*

### 4.2. Preliminary Experiment

We conducted pre-experiments on two open datasets—CIFAR-10 [80] and MNIST [81]—to verify the two hypotheses. We constructed three classifiers for each dataset (six classifiers

in total) using different classification techniques (i.e., VGG16 [82], RESNET20 [83], and AlexNet [84]), and generated a group of sub-models for each classifier. Specifically, we constructed 16 sub-models on VGG16 using all 16 hidden layers (containing both convolutional and pooling layers). Similarly, we constructed 19 sub-models on RESNET20 and seven sub-models on AlexNet. All sub-models are untrained and therefore do not have classification capability in the initial state.

We collected 10,000 samples from the test sets of the two datasets as normal samples for training. We also collected 10,000 samples from two other datasets of the same data structures, i.e., fashion-MNIST [85] and CIFAR-100 [80], as abnormal samples. These abnormal samples do not belong to any classes in the two datasets. We compared the consistency of classification results between sub-models of the two datasets when dealing with normal and abnormal samples.

Figure 4 shows the experiment results. First, both blue (normal samples) and red (abnormal samples) lines have an upward trend, indicating a gradual increase in the number of sub-models with the same classification results as the original classifier (H1). Second, blue lines are above the red lines in most cases, indicating that more sub-models of normal samples (blue lines) output the same results as the original classifiers (H2). The experiment results show that both H1 and H2 hold.

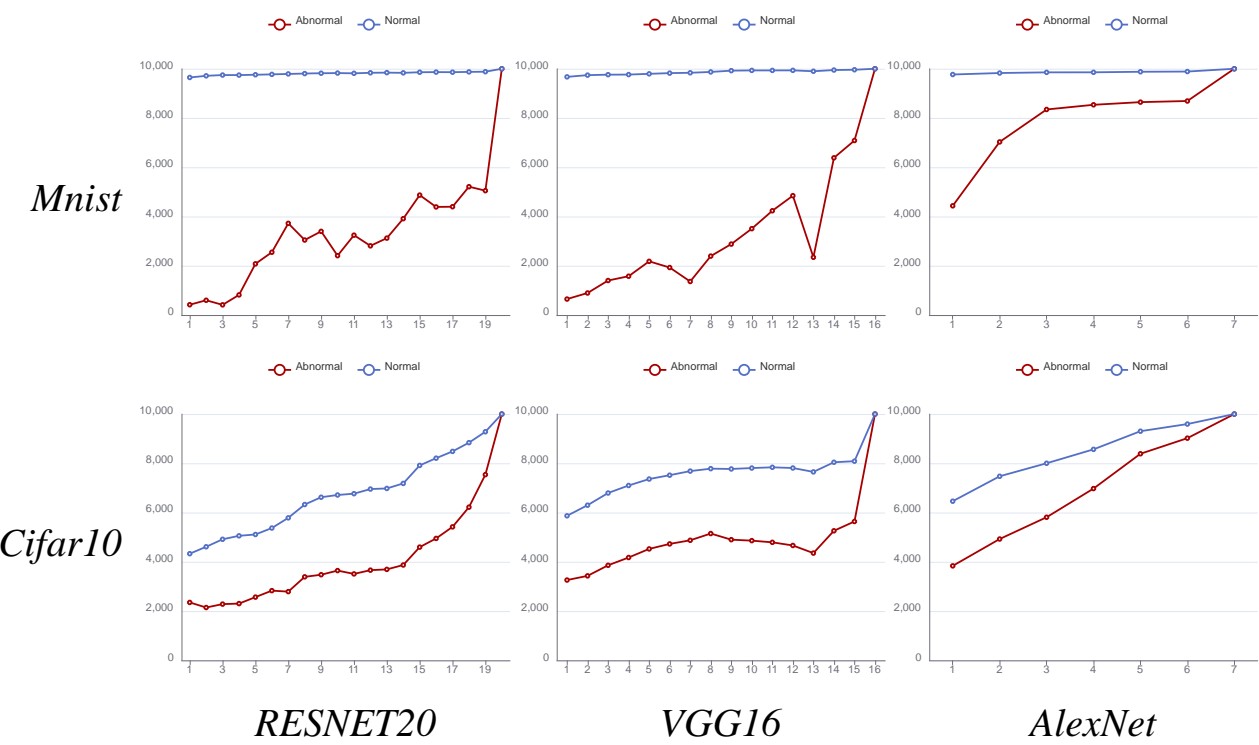

**Figure 4.** Pre-experiment results. X-axes and Y-axes indicate sub-model indexes and the numbers of sub-models with the same classification results as the original classifiers.

*4.3. Indicator*

We propose an indicator to measure the normality of the input sample according to the pre-experiment results, as follow:

$$InputNormality(x) = \frac{\sum_{i=1}^{n} Diff(f_i(x), y) \times i}{\sum_{i=1}^{n} i} \qquad (1)$$

in which $f_i(x)$ represents the output of the $i$th sub-model ($n$ sub-models in total), $y$ represents the classification result of the original classifier, and $Diff(.)$ is an operator that returns 1 if the two parameters have identical values and 0 if not. The indicator measures how many sub-models have the same classification result as the original model. The more sub-models that classify a sample into the same class as the original classifier, the higher the

indicator score of the sample. Normal samples with more sub-models outputting the same results as the original classifier thus should have higher indicator scores (H2). Moreover, we use the sub-model index as the weight of $Diff(.)$, thus reducing the effects of sub-models close to the input layer on the score, which always produces more significant fluctuations (H1). The indicator has two advantages. First, it is normalized; i.e., its value range is [0, 1]. Second, its computational complexity is low. The indicator relies on the classification results of all sub-models, each involving the calculation of only a single fully-connected output layer. The two advantages bring better usability.

## 5. Random Batch Visualization Exploration

We introduce the random batch visualization exploration (RBVE) framework from multiple aspects and then give a proof-of-concept tool that implements RBVE.

### 5.1. Framework

Figure 5 shows the RBVE framework. The analyst first initializes a PC (Step 1). After that, the data exploration begins. The analyst sets the exploring space of the target dataset by specifying a value range on each attribute (Step 2). Then, RBVE generates a batch of data queries within the specified space and generates the corresponding visualizations (Step 3). The PC identifies the pattern classes of these visualizations (Steps 4 and 5). The analyst checks the classification results to understand representative patterns in the space (Step 6). Analysts can also filter samples with lower indicator scores that are likely to contain new data patterns, and adjust the PC's settings (e.g., adding a new pattern class) according to these filtered samples (Step 7). During IDE, the analyst repeats the above process, enabling the PC to cover all identified data patterns.

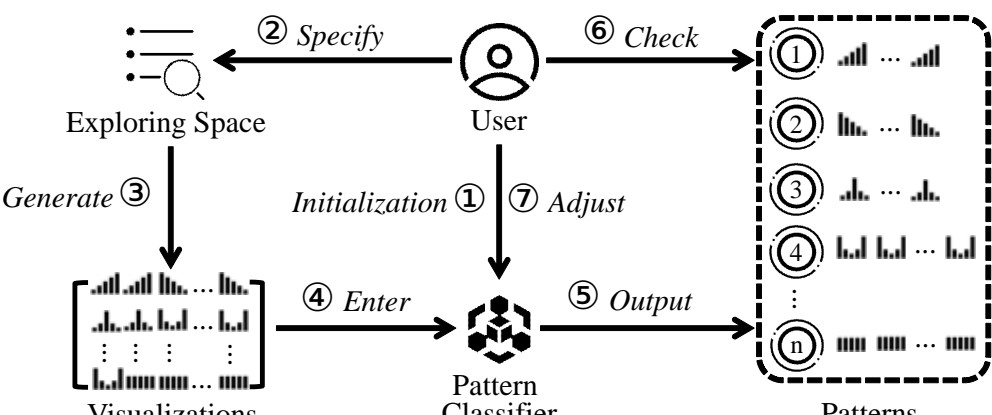

**Figure 5.** RBVE framework.

RBVE involves synthesizing data queries within a user-specified exploring space to generate a batch of visualizations. In this paper, we consider that each data query specifies value ranges on attributes to filter records, while the visualization shows the distribution of filtered records. We require each query to cover the same attribute ranges to avoid generating visualizations containing records varying wildly in scales. Specifically, we discretize the value range of each attribute into unit intervals of equal width, and each query covers a unit interval on each attribute. Each visualization thus corresponds to a cell in the specified exploring space, showing patterns of records within the cell. Figure 6 shows a visualization generation example, in which the analysts specify value ranges on three attributes, each covering two unit intervals, to generate eight visualizations ($2 \times 2 \times 2$). Note that the method of randomly generating a batch of visualizations is flexible. RBVE allows the analyst to change the strategy according to actual needs.

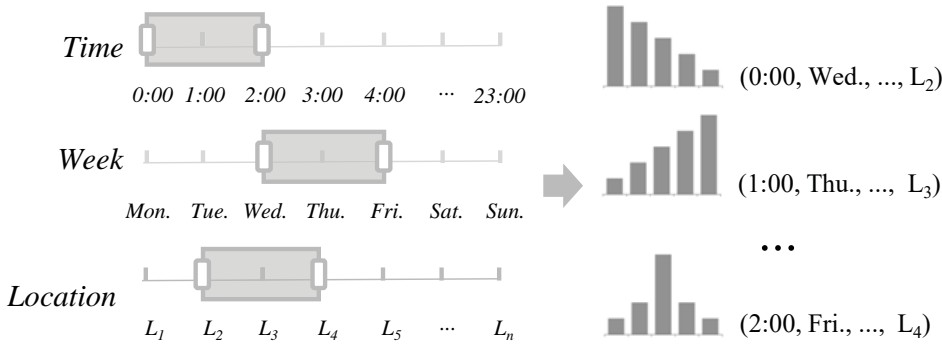

**Figure 6.** Examples of generating a batch of visualizations within specified attribute ranges.

*5.2. PC Initialization*

We believe that initializing a PC is not a challenging task. Most relevant works have given very feasible solutions [8,9]. Here, we discuss critical aspects of initializing PCs, as follows.

**Visualization Encoding:** There are two methods for encoding visualizations. First, we can process the outputs of any visualization system into images, which no longer require raw data and have better generality. Second, when the raw data are available, we can use numeric features mapped by the visual items to encode a visualization, as in Figure 7. In this case, we do not need an extra step to convert a visualization to an image. In addition, numeric features are more compact than images, which can reduce the parameters of PCs. Both formats have their strengths, and users can choose a suitable format according to the actual situation.

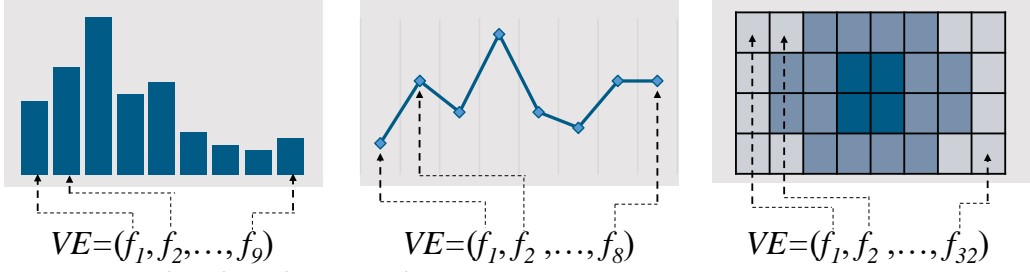

**Figure 7.** Encoding three classic visualizations as numeric vectors.

**Model Structure Choice:** The strong performance of classification techniques in existing works [9] proves their feasibility for implementing PCs. We will determine the model structure according to the visualization encoding format. When encoding each visualization as a compact numeric vector, we can implement the PC with simple model structures, such as fully connected networks. In contrast, we can use deep learning-based classifiers, such as AlexNet [84], VGG [82], GoogleNet [86], ResNet [83], and DenseNet [87], etc., for handling visualizations of the image format.

**Training Set Generation:** We give a workflow for labeling visualizations to generate the training set, as in Figure 8. We first execute queries to generate many visualizations, as in Figure 8a. Then, we cluster the visualizations into groups and use the group index as the label for each visualization [88], as in Figure 8b. Finding the optimal number of groups is unnecessary, as the analyst will gradually enrich the pattern classes during IDE. Clustering may produce results inconsistent with human perception. Therefore, we need to purify samples by removing those at the classification boundary, i.e., keeping the ideal samples for each class, as in Figure 8c.

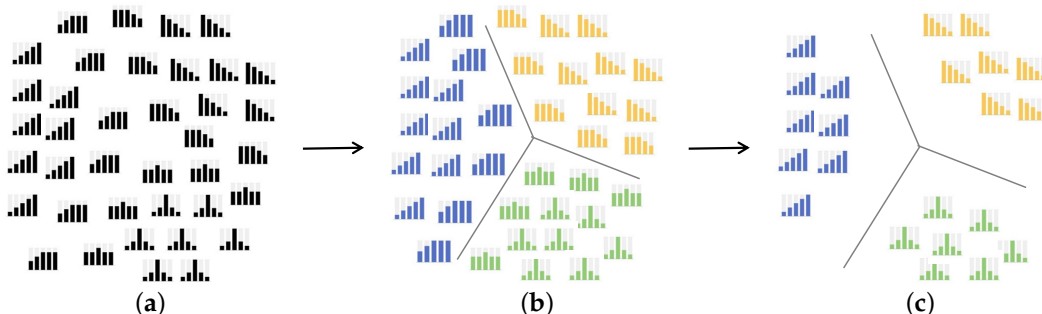

**Figure 8.** Workflow for labeling visualizations to generate the training set. The workflow integrates three steps, i.e., (**a**) sample collection, (**b**) clustering, and (**c**) pattern purification.

### 5.3. Proof-of-Concept Tool

We construct a proof-of-concept based on the RBVE framework. The tool integrates two components, i.e., the visualization explorer (VE) and the classification adjuster (CA), as in Figure 9. Please refer to the supplementary materials for videos demonstrating system usage and cases, as detailed below.

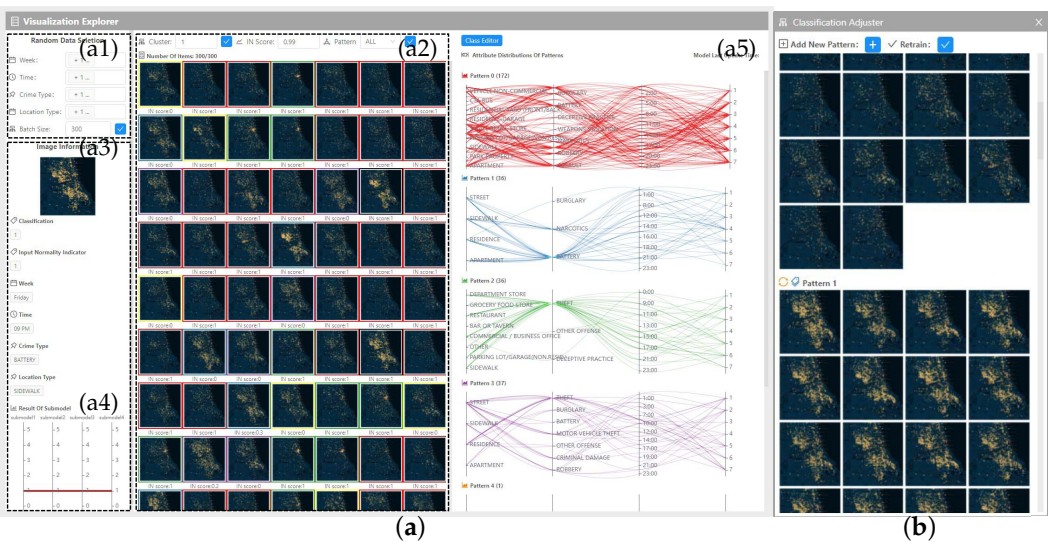

**Figure 9.** The interface of the proof-of-concept tool that integrates two components, i.e., (**a**) the visualization explorer and (**b**) the classification adjuster. (a1–a5) are the five sub-views of (**a**).

The VE controls the exploration process and shows analysis results in Figure 9a. The VE contains a query panel (a1), in which the analyst can specify a value range on each attribute. The tool can then generate a batch of visualizations within the specified attribute ranges shown in the visualization list (a2). The color of each visualization's border represents its pattern class output by the PC. Analysts can click on any visualization to see its details in an information panel (a3). The information includes the pattern class, the indicator score, attribute distribution, and classification results of parts of sub-models (a4 shows classification results of four sub-models, each corresponding to an axis). The result view (a5) includes a group of vertically aligned parallel coordinates, each corresponding to a pattern class and showing the attribute distributions of visualizations classified into this class.

The CA is for updating the PC during IDE, as in Figure 9b. The main body of the CA consists of visualizations of all pattern classes. The CA keeps the top 200 visualizations with the highest indicator scores for each pattern class. High indicator scores mean that these visualizations should have representative patterns for their respective classes. The CA will update visualizations in all pattern classes at each round based on indicator scores of existing and newly generated visualizations. During IDE, analysts interactively filter

visualizations with lower indicator scores in the visualization list (a2). Having found any new patterns in the filtered visualizations, they will click the "Add New Pattern" button in the CA to generate a new pattern class and drag visualizations with new patterns from the visualization list to the pattern class. Then analysts can click the "retrain" button to retrain the PC with visualizations on the interface as training samples.

The tool supports constructing a training set for initializing a PC. The process starts from specifying queried attribute ranges to generate a batch of visualizations in the VE's query panel (a1). These visualizations will also emerge in the visualization list (a2). Then, analysts cluster these visualizations into groups and assign the group index as the label of each visualization as a training sample. The CA will show these labeled visualizations for manual inspection. Analysts can delete visualizations that do not have ideal patterns by right-clicking on them. After obtaining the training samples, analysts can use the deep-learning libraries [89,90] to train the PC.

## 6. Quantitative Experiments

We compare our input normality indicator with common active-learning indicators, i.e., smallest margin (SM) [17], entropy-based sampling (ES) [18], vote entropy (VE) [20], and average KL divergence (KL) [22]. SM and ES are single-model-based, calculated based on the distribution of the penultimate layer. VE and KL are multiple-model-based, calculated based on the classification results of multiple weak classifiers.

Figure 10 shows the experiment results. We found that blue lines (our indicator) and green lines (SM) are above other lines in most cases, indicating that they have found more significant proportions of abnormal samples. However, the SM numbers of the found abnormal samples are far less than ours (see the numbers along the lines). Especially when the indicator score is low, SM can only find a few abnormal samples. Our indicator can accurately find many abnormal samples, even with low indicator scores (high number and percentage), indicating its better ability to identify abnormal samples.

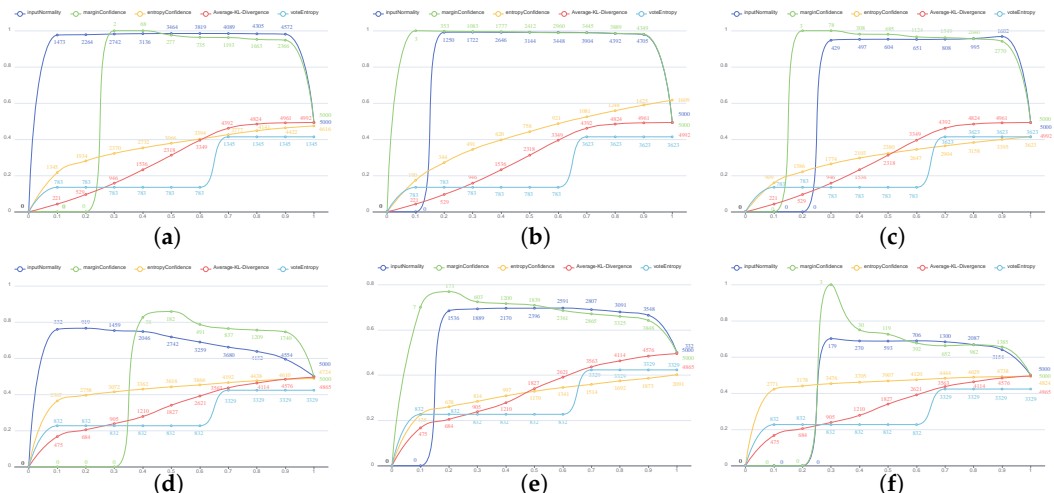

**Figure 10.** Comparing our indicator with active-learning indicators in identifying abnormal samples. The indicator score gradually increases along the X-axis in each figure, while the Y-axis represents the proportion of abnormal samples in the filtered samples. Numbers on lines indicate the number of abnormal samples found at the corresponding indicator scores. (**a**) mnist $\times$ RESNET20. (**b**) mnist $\times$ VGG16. (**c**) mnist $\times$ AlexNet. (**d**) cifar10 $\times$ RESNET20. (**e**) cifar10 $\times$ VGG16. (**f**) cifar10 $\times$ AlexNet.

## 7. User Study

We further conducted a user study to verify the feasibility of the RBVE framework, detailed below:

### 7.1. Experiment Design

**Dataset:** We utilized the Chicago Criminal Dataset (https://www.kaggle.com/chicago/chicago-crime, accessed on 1 March 2022), which includes 7,941,282 criminal records that happened between 2001 and 2017. We chose four attributes, i.e., day-of-week (7), hour-of-day (24), crime type (35), and location type (170), to form an exploring space with 999,600 cells ($7 \times 24 \times 35 \times 170$). We thus can generate the same number of visualizations as total cells, each showing the spatial distribution of crime records within a cell, as in Figure 11.

**PC initialization**: We randomly generated 20,000 visualizations and clustered them in four classes as the training set. We saved each visualization as an image of $445 \times 470$ pixels and implemented the PC as a convolutional neural network to handle them. The network integrates six convolutional layers, each coupled with max pooling, and two fully-connected layers. We integrated the PC into the proof-of-concept tool to form the experiment environment.

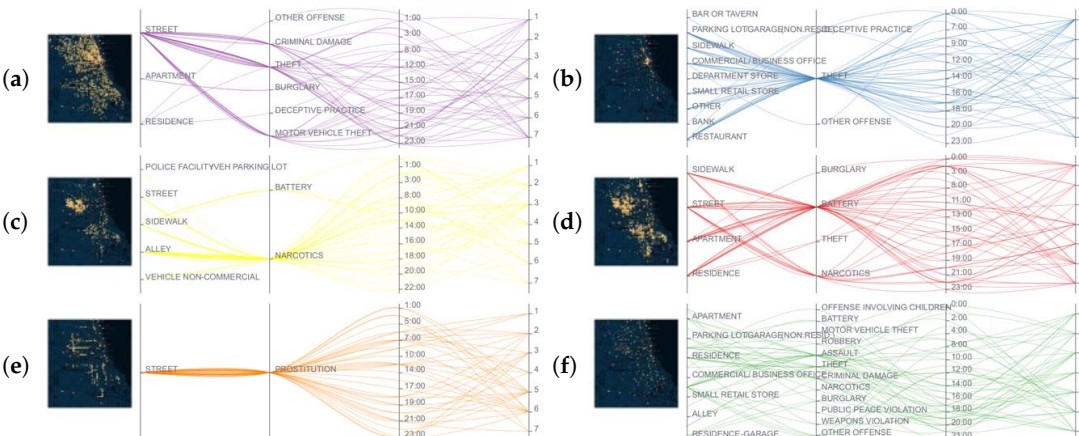

**Figure 11.** (**a–f**) Six representative patterns identified using our approach. .

**Participants**: We recruited 20 participants (12 males and 8 females, aged 22–25). All participants had some knowledge of visual analytics, but none had participated in the project before. We asked each participant to use the proof-of-concept tool to find as many spatial patterns as possible within the exploring space.

**Process**: There was a training session at the beginning of the user study. During the training session, we introduced the general idea, the principle of the indicator, and the usage of the proof-of-concept tool for the participants. We set an experimenter to answer the questions raised by the participants. We ensured that all participants understood our research and used the proof-of-concept tool proficiently. We set up screen-recording software to record participants' operations during the experiment. The participant could terminate the experiment if he (or she) could no longer find more patterns. After the experiment, each participant filled out a questionnaire to score four aspects of our method using the 7-point Likert scale (7 is the best and 1 is the worst). The four aspects are (1) how quickly they can find data patterns using the tool (efficiency), (2) how accurately the tool can find abnormal samples (effectiveness), (3) whether they can learn the usage of the tool quickly (easy-of-use), and (4) whether they would like to use the PC in their research (favorability). We also encouraged participants to write down their subjective comments (both positive and negative) in the questionnaires.

### 7.2. Representative Patterns

Figure 11 shows six representative patterns identified by most participants. We found the spatial and temporal distributions of thefts occurring in streets and apartments are more dispersive (Figure 11a), while those occurring in commercial facilities, such as stores, restaurants, etc., are mainly concentrated in a small region to the northeast (Figure 11b). Violent crimes, such as battery, narcotics, criminal damages, etc., mainly occur in a large region to the northwest (Figure 11c). At the same time, battery crimes

also happen in the south of the city (Figure 11d). Figure 11e shows the spatiotemporal patterns of prostitution activities that occur primarily on the streets. The spatial distribution in Figure 11f is relatively irregular and does not correspond to a specific criminal type. It, however, represents a pattern class of fewer crimes occurring.

### 7.3. Objective Performance

Figure 12 shows participants' performance in the experiments. Most participants found over nine patterns (Figure 12a) with a few queries (Figure 12b). Moreover, most participants (except for two) could complete the experiment within 30 min (Figure 12c).

**Insight 1**: The strong performance shows the effectiveness and usability of RBVE in automatically identifying patterns of a batch of visualizations generated randomly.

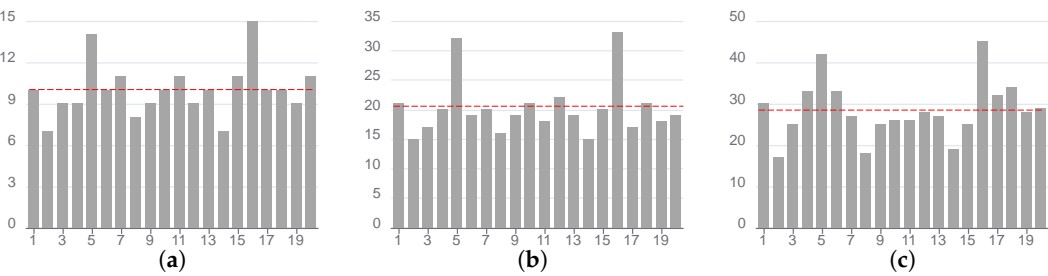

**Figure 12.** Objective performance of participants in experiments. The red line represents the average value of each variable. (**a**) Number of new patterns. (**b**) Number of queries. (**c**) Completion time.

Two participants took more than 40 minutes (Figure 12c) to complete the experiments. By observing their screen recording videos, we found that they conducted more queries and identified more pattern classes from the filtered visualizations with low indicator scores. More frequent retraining of the PC led to longer experiment time. We carefully checked their identified patterns and found that the differences between these classes are not very significant.

**Insight 2**: New patterns determined by PCs are sometimes different from human perceptions. Participants with strict (or relaxed) criteria for pattern classes may merge (or separate) patterns that contain subtle differences, thus creating fewer (or more) pattern classes. Although the indicator can improve the automation of identifying new patterns by filtering a few candidates, it still cannot replace human decision. We will study how to further reduce human intervention in determining new pattern classes in the future.

The number of queries (Figure 12b) is higher than that of patterns (Figure 12a), indicating it is unnecessary to update the PC in each query. Moreover, we found that PC retraining mainly occurred in the early stage of each experiment and gradually decreased in frequency as the number of patterns included in it increased, as shown in Figure 13.

**Insight 3**: The above phenomena further illustrates the effectiveness of the PC in improving IDE efficiency. The RBVE framework can avoid frequently updating PCs. As the number of included pattern classes increases, the PC will pick fewer samples with low indicators, gradually decreasing its update frequency. In the later stage of the experiment, the PC has recognized most data patterns, thus making the IDE automated without any updates.

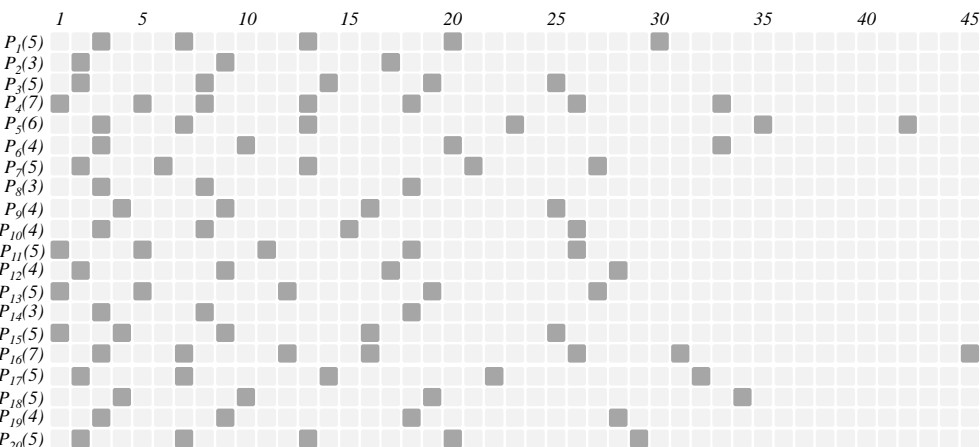

**Figure 13.** Number and time of PC retrainings for each participant.

### 7.4. Subjective Scores

Figure 14 shows participants' scoring results. Almost all participants gave the highest scores on efficiency. Participants agreed that the tool could calculate visualizations' indicator scores in real-time. Moreover, they considered exploring a whole exploring space as helpful in improving IDE efficiency.

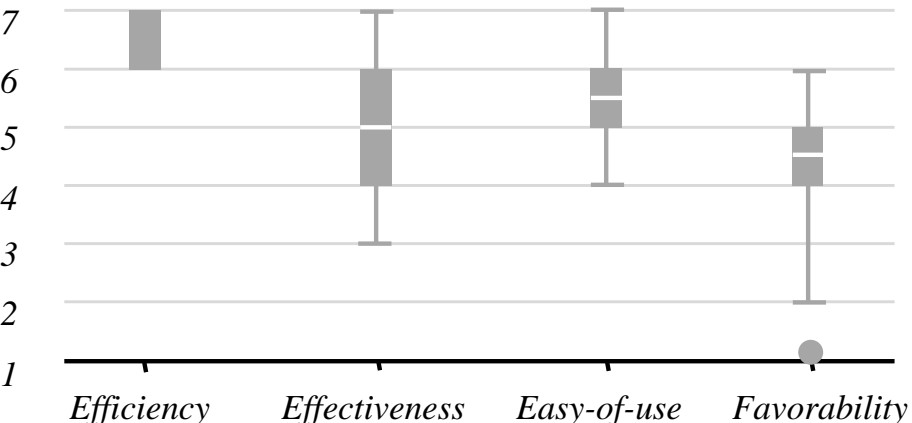

**Figure 14.** Scoring results on four aspects.

**Insight 4**: The higher efficiency scores derive from the low computational complexity of the indicator. The RBVE framework thus enables the analyst to quickly identify abnormal samples from a batch of visualizations generated on the fly.

Effectiveness receives relatively high scores. All participants acknowledged that most visualizations with low indicator scores contain new patterns. However, they also found a few exceptions with relatively high scores, but new patterns, which affects scores of this aspect. Moreover, two participants said they could not use a constant threshold during the data exploration. They sometimes had to change the threshold to find abnormal samples.

**Insight 5**: Our method can guarantee that most low-scoring visualizations contain new patterns. However, a few exceptions are inevitable due to the inherent uncertainty of deep learning. The RBVE framework thus depends on the analyst to choose new patterns from visualizations with low indicator scores. The positive aspect is that the retraining frequency will gradually decrease as patterns covered by the PC increase, as in Figure 13.

**Insight 6**: It is impossible to set a uniform threshold that is applicable for all cases. Our indicator can find a significant proportion of abnormal samples at lower values (e.g., <0.2; see Figure 10). However, after the PC has identified most data patterns, almost no low-scoring samples exist. At that time, analysts need to interactively increase the threshold to find abnormal samples in a broader range.

Scores of ease-of-use are high, and the score distribution is more concentrated. Participants agreed that they could easily understand the RBVE framework and grasp the usage of the tool. Several participants also pointed out a few limitations of the proof-of-concept tool and gave us suggestions to improve the system. For example, they suggested including more PCs to enable the analysts to explore multiple kinds of data patterns simultaneously. On the other hand, a few participants felt that it was inconvenient to start external programs to modify the PC structure and hoped the tool could support modifying the PC structure interactively.

**Insight 7**: The tool only integrates core components to verify the feasibility of the RBVE framework. According to the experiment results, we believe implementing the functions suggested by the participants is feasible. For example, we can integrate more PCs into the tool. Analysts thus can explore multiple types of data patterns at the same time. High classification efficiency and low computation complexity ensure that the tool with multiple PCs can respond to user interactions in real-time. We plan to implement the functions suggested by the participants in the future.

Participants' scores on favorability varied widely. Some of the participants agreed that our method could improve the IDE efficiency and scored this aspect high. However, participants who are more familiar with visual analytics pointed out that our approach is very different from the relevant research on authoring visualizations [3,4,70]. They do not know whether our method is better.

**Insight 8**: Our method is not an improvement of existing methods that have already shown their usability in the demonstrated cases. Instead, we focus on a new IDE problem, i.e., identifying patterns from the generated visualizations, which has not received much attention. It is possible to use PCs and other techniques together in IDE.

## 8. Conclusions

This paper generalizes the concept of the pattern classifier based on a few newly-emerging cases of applying supervised classifiers to identify patterns from visualizations. We target a practical data exploration scenario wherein pattern classes cannot be determined in advance and will change dynamically during the exploration. We identify the main technical challenges of applying pattern classifiers in the scenario and give two methods, i.e., an active-learning indicator and a visualization framework, to resolve them. We consider pattern classifiers and existing automatic visualization techniques as technically complementary, automating different stages of interactive data exploration, and can work together to improve the exploration efficiency. Our study points to a new direction for automatic visualization and may facilitate the emergence of other similar studies.

In the future, we plan to improve the method in following aspects. First, we will test the active-learning indicator on more classic classifiers to verify its effectiveness. Second, we would like to refine this proof-of-concept tool according to the suggestions proposed by the participants in the user study, such as integrating multiple pattern classifiers to identify different types of patterns and enabling users to modify classifier structure in the interface.

**Supplementary Materials:** The following supporting information can be downloaded at: https://www.mdpi.com/article/10.3390/app122211386/s1, Video S1: PC initialization, Video S2: PC application.

**Author Contributions:** Methodology, J.L.; Software, W.H.; Validation, H.T.; Visualization, H.T.; Writing—original draft, W.H.; Writing—review & editing, J.L. All authors have read and agreed to the published version of the manuscript.

**Funding:** This work is supported by the NSFC project (61972278) and the Natural Science Foundation of Tianjin (20JCQNJC01620).

**Data Availability Statement:** Not applicable.

**Conflicts of Interest:** The authors declare no conflict of interest.

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
