# Peer review of "Active Pattern Classification for Automatic Visual Exploration of Multi-Dimensional Data"

_applsci, doi:10.3390/app122211386_

Round 1

Reviewer 1 Report

This work introduces an active learning-based multidimensional data pattern classification pipeline for improving the effectiveness and efficiency of interactive data pattern exploration. It has two main contributions: first, it introduces an active learning metric to measure the normality of data instances. Second, the entire pipeline uses an interactive design scheme that allows users to find patterns progressively in a multidimensional dataset.

Overall, I think this is an interesting work demonstrating visual analytics's power in pattern exploration, and the design of the metrics and models seems good. However, I also note some issues in the current version of the manuscript.

Strength:

1. The research topic is quite interesting. To extract useful patterns from large datasets, it is essential to add automated components to the visualization workflow. This topic reveals this promising research direction.

2. Technically, this work lies between fully automated class incremental learning and fully interactive visual exploration. It provides a "semi-automatic" approach to exploit both normative measurements (automatic) and active labeling (interactive).

3. This study presents a well-designed interactive interface.

Weakness:

1. Expression issues

- The RBVE framework proposed in this study may be a new contribution to IDE. However, the current presentation only describes the working mechanism of the new framework, but lacks detailed design rationale (i.e., why do we need such a framework? its unique advantages, and application scenarios, etc.). 

- It is positive that this study tested the proposed active learning metrics through a pre-experiment. However, the authors lack the necessary description of the details of the experiment, which limits the validity of the experiment to some extent.

2. Related work to be added

- The use of normality measurement to detect new class instances belongs to a traditional area of research in machine learning - class incremental learning [1,2,3] (this is only a partial list of recent works). Therefore, it is necessary to discuss representative works of this type of approach.

[1] Class-incremental learning via deep model consolidation.

[2] Il2m: Class incremental learning with dual memory.

[3] Distilling causal effect of data in class-incremental learning.

- In addition, subspaces can be viewed as dimensional snapshots of high-dimensional data. However, there is no discussion of visualization work on subspace analysis in this manuscript. Related works such as (but there are more).

[4] Pattern Trails: visual analysis of pattern transformations in subspaces.

[5] Optimal projection sets for high-dimensional data.

3. Writing issues

I find several grammatical errors and typos, as well as sentences with unclear meanings. 

Sec 1: The remain (remaining?) part of the paper is organized...

Sec 2: Many tools, such as..., also depends (depend?) on indicators to…

Fig 4 caption: please be consistent with 'n' or 'N' in the figure and caption.

Sec 5: The analyst first initializes a PC (Step 1, Sections 3.1). -> Section 3.1?

Author Response

Point 1: Expression issues.
Response 1: We added the design rationale of the framework in Section 1 (see the fifth paragraph in Section 1). We also detailed the pre-experiments in Section 4 (see the first paragraph in Subsection 4.2).

Point 2: Related work to be added.
Response 2: We have added all references recommended by the reviewer (see the second paragraph in Section 1 and the fourth paragraph in Subsection 2.2).

Point 3: Writing issues.
Response 3: We thoroughly revised the text and figures to eliminate grammatical errors and typos.

Reviewer 2 Report

Article is written with proved PC-based data exploration approach. Include more keywords. 

Author Response

Point 1: Article is written with proved PC-based data exploration approach. Include more keywords. 

Response 1: We added more relevant keywords. Now the keywords include interactive data exploration, automatic visualization, neural network, machine learning, active learning, visual analytics.

Reviewer 3 Report

Overall, this is an  interesting paper and a useful contribution to the literature. I have some suggestions that would make this paper more robust.

1. Section 3 on page 4 - Problem statement is not clear, could be improved. Provide concise description of the issue that needs to be addressed; Should be short and clear.

2. Challenges could be discussed before section 3 /Problem statement to make more sense.

Author Response

Point 1: Section 3 on page 4 - Problem statement is not clear, could be improved. Provide concise description of the issue that needs to be addressed; Should be short and clear. Challenges could be discussed before section 3 /Problem statement to make more sense.

Response 1: We added the application scenario of our approach in the problem statement section. We then summarize two technical challenges for achieving the scenario as the basis of the two main contributions (Sections 4 and 5). The content of PC initialization is moved to Section 5, as it is more related to the system implementation.